# Free-space optical communications at 4 Gbit/s data rate with a terahertz laser
Jayaprasath Elumalai [1] ✉, Mohammed Salih [1], Martyn Fice [2], Adam Brown[1], Lianhe Li [1], Edmund H. Linfield [1], Alexander Valavanis [1], Alwyn J. Seeds[2], Alexander Giles Davies [1] & Joshua R. Freeman [1] ✉

Terahertz-frequency (THz) carrier waves in free-space optical (FSO) communications offer the potential for > 1 Tbit/s data rates and stable latency. They offer wider bandwidths than available in the microwave region, together with reduced scattering and relaxed pointing requirements compared with visible and near-infrared regions. However, 1–10 THz FSO communications systems have thus far been limited to data rates several orders of magnitude lower than those of infrared systems. This work describes an experimental demonstration of multi-gigabit-per-second FSO communication using a THz quantum cascade laser (QCL), opening a new frontier for next-generation wireless communications. The FSO communication system consists of a 2.4 THz QCL source as the transmitter and a room-temperature Schottky barrier diode detector as the receiver. By directly modulating the terahertz QCL, we achieved non-return-to-zero on-off keying (NRZ-OOK) with a transmission rate of up to 4 Gbit/s. We evaluated the performance of the communication link by analyzing the bit error rate (BER) of the demodulated signal at the receiver while examining its relation to received optical power, QCL modulation power, and various bias points. Our work establishes the foundation for high-speed optical wireless communication based on terahertz QCL technology systems.

Connected devices and cloud services play an increasingly important role in our lives, and the demands on network bandwidth, particularly in wireless networks, are predicted to continue to increase rapidly. These requirements are driven by a range of emerging applications such as extended reality, smart homes and businesses, autonomous systems, brain-computer interfaces, telemedicine, cloud-based AI services, and satellite communications[1–7]. These requirements drive the development of communication systems in the radio and microwave frequency bands towards higher frequencies to increase bandwidth in so-called 6G communication networks. The drive for increased wireless bandwidth also drives the development of wireless communication networks in the visible and near-infrared regions of the electromagnetic spectrum, where the potential bandwidth is huge, owing to the very high carrier frequencies. Free-space optical (FSO) links have demonstrated data rates reaching several hundred gigabits per second (Gbit/s) to terabits per second (Tbit/s), coupled with low latency (<1 ms)[8–10]. The "optical links" refer to free-space laser-based links, extended here to the terahertz frequency band. FSO links are expected to play an essential role in future network infrastructure, combining radio-frequency and optical wireless communication. Radio-frequency (RF) and

FSO links connect various sectors, from data centers to smart farms. Figure 1 highlights seamless optical wireless communications and Internet of Things (IoT) devices in a connected smart ecosystem[11]. There are also proposals to use FSO links for ultra-low-latency rack-to-rack connections within data centers. However, FSO links in the visible and near-infrared spectrum do face challenges. These stem from the very short wavelengths, which are more susceptible to Rayleigh scattering in the atmosphere, which scales inversely with the fourth power of the wavelength ($\lambda^{-4}$). Shorter wavelengths also require very stringent beam-pointing alignment criteria to maintain acceptable link loss as frequency increases.

These fundamental challenges have motivated the investigation of intermediate frequencies between the microwave and visible ranges, which have the potential to balance lower link loss with visible light while maintaining the capability for very high bandwidths and the intrinsic security of highly directional communications. Recent studies show communication links achieving speeds exceeding several Gbit/s at carrier frequencies from 30 to 65 THz (4.65 to 9.6 μm wavelength) utilizing a mid-infrared quantum cascade laser (QCL), intersubband modulator, and intersubband detector[12–17]. More recently, the terahertz (THz) frequency band, from

[1]School of Electronic and Electrical Engineering, University of Leeds, Leeds, UK. [2]Department of Electronic and Electrical Engineering, University College London, London, UK. ✉e-mail: j.elumalai@leeds.ac.uk; j.r.freeman@leeds.ac.uk

**Fig. 1 | THz optical wireless communication networks.** A conceptual diagram of the future network applications utilizing THz optical wireless communication links across various sectors, such as inter-satellite communication, inter-rack communications in data centers, and smart homes.

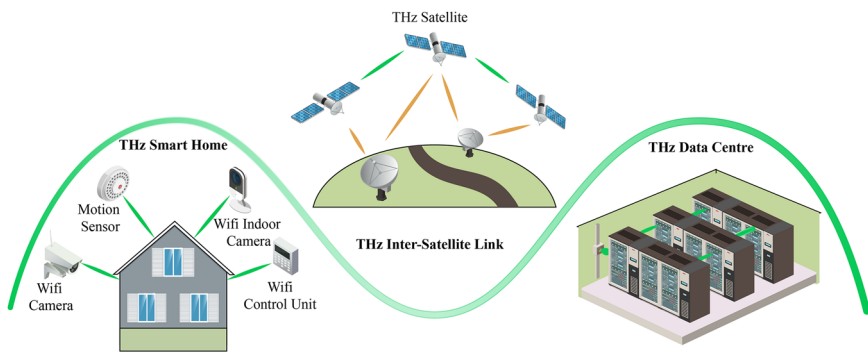

approximately 0.3–10 THz, has emerged as a new frontier, offering the promise of data rates exceeding the Tbit/s limit[18]. Furthermore, the longer wavelength of THz radiation compared to infrared radiation and visible light provides inherent robustness against Rayleigh scattering by atmospheric particulates such as dust and smoke. The THz spectrum faces challenges due to atmospheric absorption, which limits transmission distance across much of the band. However, there are transmission windows, notably at 0.3, 0.6, 2.1, 2.54, 3.4, and 4.9 THz, where excess loss due to atmospheric absorption can be less than 1 dB/m[19]. As such, these frequency bands are suitable for short-range terrestrial communication links[20,21]. Furthermore, atmospheric absorption in THz FSO communication becomes negligible for inter-satellite links, where the absence of atmospheric interference allows for high-speed THz FSO communication (Fig. 1). Coupled with the reduced pointing accuracy required for THz FSO communication compared to optical or infrared radiation, this makes a THz carrier an attractive option for satellite-based communication systems[22].

The lower end of the THz range (0.3–0.6 THz) has been extensively explored for wireless communication. This spectral region is accessible using electronics-based devices, and various wireless communication links have been demonstrated[23–30]. However, the 1–10 THz frequency band remains largely unexplored. This is despite intensive work on sources, modulators, and receivers in this frequency band, improving output powers, modulation bandwidths, and detector sensitivities; recent reviews of these devices can be found in references[31–35]. The most promising THz sources in this range include high-power QCLs that operate in the 2–5 THz range, exhibit fast dynamics[36], and can be operated in Peltier coolers[37]. Modulators have been developed based on graphene and plasmonic structures positioned on top of semiconductor heterostructures and are capable of modulation frequencies up to 1 GHz[38,39]. There are also a variety of fast detectors available, including cooled quantum well photodetectors[40], photodetectors based on silicon MOSFETs[41,42], graphene[43], and Schottky barrier diodes[44,45].

Despite these technological advancements, QCL-based FSO systems have remained limited to impractically low data rates, negating many of the potential advantages of operating in this band. The fastest THz FSO system to date achieved a data rate of 20 Mbit/s using a cryogenically cooled quantum well detector[46]. A more recent FSO system using a cryocooler for QCL operation and a room-temperature graphene-based receiver provides a data rate of 1 Mbit/s over a 5 m path length and 115 kbit/s over a 7.5 m path length[47]. Despite recent advances in device development, establishing a THz FSO communication link capable of multi-gigabit-per-second data rates remains a significant challenge. This requires a careful balance of source power, modulation efficiency, and detector sensitivity, all integrated into a system that functions reliably under practical conditions.

In this work, we combine some of the most promising technologies in this frequency band and demonstrate a THz FSO communication link with a 4 Gbit/s data rate using a QCL transmitter and a room-temperature Schottky barrier diode as a receiver. The THz QCL carrier frequency is dominated by a 2.4 THz mode and is operated in a closed-cycle cryocooler. Rather than employing a separate modulator, we show that it is possible to modulate the QCL emission directly over a wide bandwidth without significant distortion or excessive modulation power requirements. We investigate encoding data in NRZ-OOK and pulse-amplitude-modulation-4 (PAM-4) format signals at a free-space transmission distance of 0.5 m. The performance of the communication system is evaluated as a function of the QCL optical power for different data rates. We demonstrate that the transmission of NRZ-OOK signals at data rates approximately 200 times higher than those previously reported in refs. 46,47 is achievable. Our demonstration establishes the foundation for developing high-speed optical wireless communication systems based on THz QCLs. By demonstrating data transmission rates in the gigabit-per-second range, this work offers a promising solution for next-generation communication networks, particularly in satellite communications and inter-rack communications within data centers.

## Results

### THz FSO communication link

The experimental arrangement of a THz FSO communication link is schematically depicted in Fig. 2, with a transmitter comprising a QCL, a collimating optic, an FSO channel, and a Schottky barrier diode as a receiver. The transmitter employs an arbitrary waveform generator (AWG) with a sampling rate of 12 GSa/s to generate the NRZ-OOK and PAM-4 format data signals. The AWG output is connected to a 27 dB gain RF amplifier and a bias-tee to modulate the QCL. The THz QCL employs a "metal–metal" waveguide Fabry–Pérot cavity with dimensions of $(2000 \times 70)$ $\mu m^2$ emitting at 2.4 THz (dominant longitudinal mode) with weaker side-modes around 2.6 THz. Although advances in QCL design have enabled pulsed-mode operation with thermoelectric cooling[48], continuous-wave operation remains limited to $\lesssim 130$ K[49]. As such, the QCL in this work was mounted within a closed-cycle cryostat system at 19 K. Further technical details on QCL fabrication and packaging are provided in the "Methods" section.

The spectrogram shown in Fig. 3a illustrates the emission characteristics of the THz QCL as a function of drive current. We measured the QCL emission frequencies by mixing the output from the QCL with the 4th harmonic of the output from an amplifier-multiplier chain (SXG-M WM-380, Virginia Diodes) on the Schottky barrier diode mixer. An electrical spectrum analyzer (Rohde & Schwarz FSW3030) recorded the intermediate frequency (IF) signal after the +15 dB low-noise amplifier. To retrieve the complete QCL spectrum, we scanned the amplifier-multiplier chain frequency at several QCL drive currents. The plot shows the power of each QCL mode as a function of drive current over a range from 180 to 300 mA. The emission is dominated by a mode at 2.4 THz with other much weaker modes at frequencies from 2.38 to 2.64 THz. The amplified IF power associated with the dominant mode varies from −64.4 to −47.5 dBm as the drive current increases from 180 to 300 mA. The next strongest mode is at a frequency of 2.63 THz, with an associated amplified IF power 15 dB lower than that of the 2.4 THz mode. Further details are given in Fig. S2 in the Supplementary Note 1.

The relationship between THz power, current, and terminal voltage ($L$–$I$–$V$ characteristics) of the QCL device under continuous-wave (CW) operation is shown in Fig. 3b, and the threshold current ($I_{th}$) of the device is

**Fig. 2 | Principle of operation.** Experimental arrangements of the THz FSO communication system. The data is generated offline and loaded onto the AWG. The transmitter (blue box) includes an AWG for generating the NRZ-OOK and PAM-4 signals, a 27 dB gain RF amplifier, a QCL driven by a bias-tee to combine the DC and RF components, and a 90° off-axis parabolic mirror to collimate the beam. The modulated THz signal travels through free-space to the receiver section (red box) and is focused onto a Schottky barrier diode 4th harmonic mixer used as an envelope detector to detect the signal. The received signal passes through another bias-tee, followed by a 15 dB gain low-noise amplifier before being captured on an oscilloscope (OSC). The OSC converts the signal into digital samples, which are then processed offline to evaluate the signal performance.

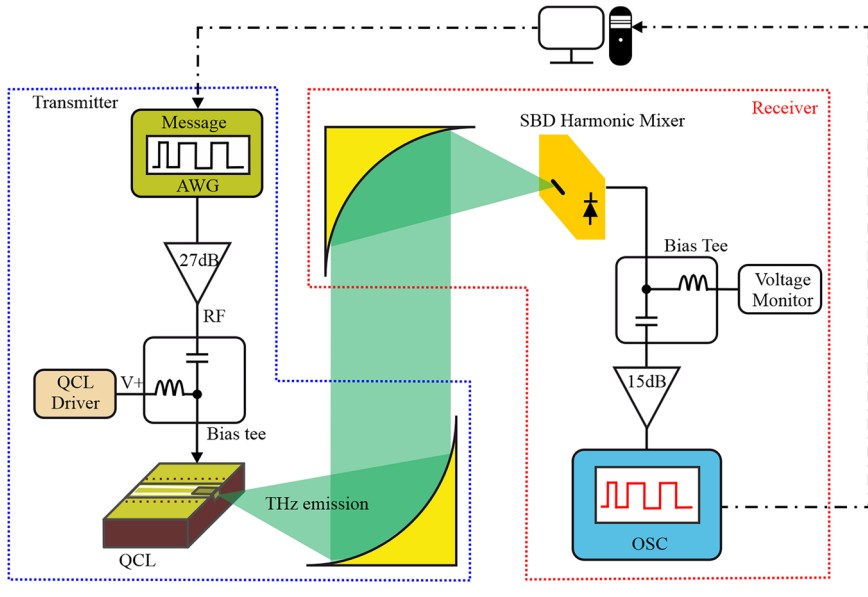

175 mA. Above threshold, the THz power does not rise linearly with current. In CW operation, the electrical power $V \cdot I$ is dissipated as heat in the active region, thereby increasing the lattice temperature, shifting the gain spectrum, and raising internal losses. This reduces the differential efficiency, giving a sub–linear $L$–$I$ slope. Small steps in the $L$–$I$ red trace arise when the longitudinal Fabry-Pérot modes move relative to the gain peak and individual modes cross threshold. Mode competition also contributes to the sub-linear $L$–$I$ characteristic and to the discrete steps seen in Fig. 3b. Over the range $I = 250–350$ mA the differential efficiency is approximately constant (gray band "modulation range" in Fig. 3b). Above $I \gtrsim 360$ mA due to an increased self–heating and field–domain formation reduce the gain hence the output power peaks and then rolls over[50–54].

We set the QCL DC current $I = 300$ mA ($1.7 I_{\text{th}}$), resulting in a primary emission frequency of 2.4 THz in our investigation and producing an optical power of 75 μW without additional modulation at an operating temperature of 19 K. In the $L$–$I$–$V$ plot, the gray shaded portion (labeled "modulation range") identifies the operational window determined by the modulation power ($P_{\text{mod}}$), where the applied 19 dBm RF power corresponds to 48 mA rms current in QCL operation. This was deduced by $I_{\text{mA}} = \sqrt{(10^{P_{\text{dBm}}/10})/(1000 \times Z_{\text{QCL}})}$ where $Z_{\text{QCL}}$, the measured RF impedance of the QCL, is 35 Ω. Further details are given in Supplementary Note 3. In this modulation range, the QCL output power is approximately linear with the drive current ($I$), enabling stable, controllable modulation of the emitted radiation. The AWG-generated baseband message is encoded onto the THz QCL emission using a 12 GHz bandwidth bias-tee. A 90° off-axis parabolic mirror with a focal length of 50.8 mm is used to collimate the beam.

Figure 3c shows the frequency response characteristics of the full system, comprising all the electrical and optoelectronic devices in the setup (bias-tees, QCL, Schottky barrier diode, and amplifiers). We use a vector network analyzer (Agilent Technologies, E8364B) to measure the full system's bandwidth by injecting a 0 dBm electrical signal into the QCL side of the bias-tee. The red curve in Fig. 3c shows the end-to-end S21 amplitude response of the system, while the gray trace shows the transmission of the same system when the QCL is turned off. The system exhibits a 3-dB bandwidth of approximately 5 GHz. The reflection coefficient ($S_{11}$) for the system is shown in Fig. S4 of Supplementary Note 3.

At the receiver, a second 90° parabolic mirror with a focal length of 152.4 mm focuses the modulated THz signal onto a high-speed SBD, which serves as an envelope detector for optical-to-electrical conversion. For the SBD used here, we measure a responsivity of 31 V/W and noise equivalent

power of 35 pW/Hz$^{0.5}$ (further details are given in Supplementary Note 2). The Schottky barrier diode is connected to a bias-tee with a bandwidth of 26.5 GHz to separate the DC and encoded message components of the detected signal. The demodulated message is amplified using a low-noise amplifier with +15 dB gain to provide sufficient voltage swing for the digitizer and to overcome the ADC front-end noise. A 20 GSa/s, four-channel digital oscilloscope captures the demodulated signal to analyze the transmitted NRZ-OOK and PAM-4 formats. The total paraxial propagation distance of the signal along the free-space link is 0.5 m.

In our experiment, we extensively studied the transmission performance of a THz FSO communication system using the NRZ-OOK modulation format at various Gbit/s data rates. Furthermore, we extended our investigation to test the system with the PAM-4 signal format at a data rate of 2 Gbit/s.

## Analysis of the THz FSO communication system

Having established the THz FSO communication system, we analyze its transmission performance in detail at a data rate of 1 Gbit/s. The NRZ-OOK message signal modulates the THz QCL around the 300 mA operating point and is transmitted to the receiver. The NRZ-OOK modulation scheme encodes the message by representing binary "1" and "0" bits. Channel 1 of the oscilloscope directly captures the transmitted message signal from the AWG using a 3-dB RF power splitter, while a second channel records the message demodulated by the SBD, which acts as a direct detector, after passing through the FSO communication system. The transmitted and demodulated signals are shown in Fig. 4a, b, respectively, for a message consisting of a pseudorandom binary sequence (PRBS) of length $2^{15}-1$. As is evident in Fig. 4b, the demodulated signal maintains sufficient accuracy for reliable data decoding, with minimal bit errors observed during the analysis.

Figure 4c shows the power spectrum of the demodulated signal (red trace) captured by an electrical spectrum analyzer. The minima in the spectrum at 1, 2, 3, and 4 GHz are the most significant features, indicating NRZ-OOK modulation at 1 Gbit/s (1 Gbaud). The sharp peaks at 1, 2, and 3 GHz reflect distortion in the received signal. The NRZ-OOK signal is a form of rectangular pulse modulation, which inherently produces higher-order harmonics due to the sharp transitions between the "0" and "1" bits. The broad peaks between the minima decrease in optical power with increasing frequency, reflecting the pulse shape. Due to finite rise and fall times, their amplitudes decrease more rapidly than those of perfectly square pulses. The spectrum is captured using a low-noise-level

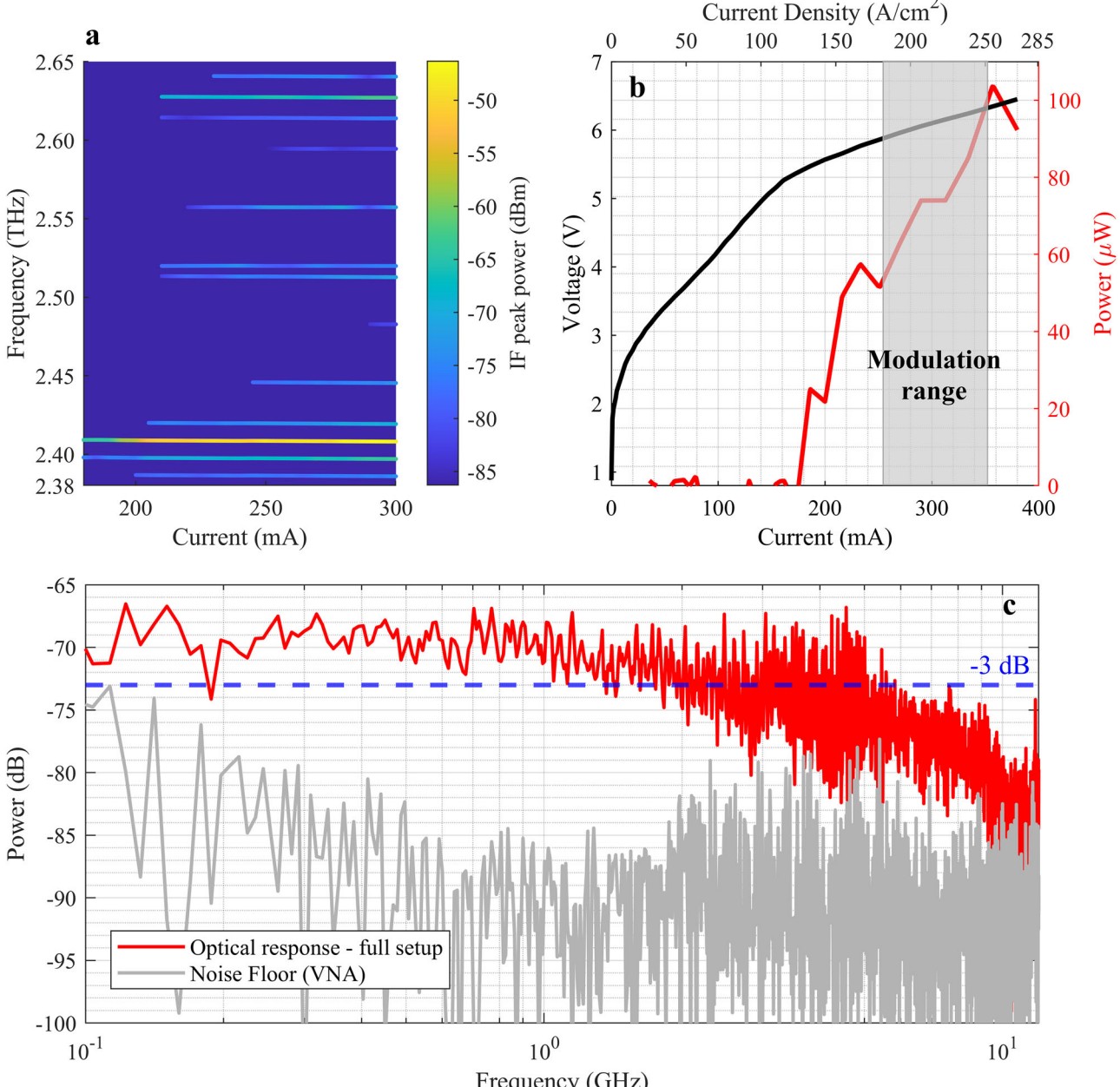

**Fig. 3 | QCL device characteristics and measured system response. a** Spectrogram of THz QCL emission as a function of drive current. The color scale indicates the amplified mixer-intermediate frequency (IF) power derived from each QCL mode. **b** CW $L$-$I$-$V$ characteristics of the QCL in the absence of modulation measured at a temperature 19 K. Red: THz power vs current showing sub-linear rise with mode-related steps and a peak before roll-over; Black: voltage vs current. Gray band marks the current range used for direct modulation, where the differential efficiency is approximately constant. **c** The full system's end-to-end (QCL, Schottky barrier diode detector, bias-tees, and amplifiers) S21 amplitude response, including the cascaded frequency response of all the electrical and optoelectronic devices in the setup, measured with a vector network analyzer (VNA). The red trace shows the amplitude response of the complete system. The gray trace shows the system's noise floor, recorded under the same configuration with the QCL off.

electrical spectrum analyzer. The gray trace in Fig. 4c shows the response of the system when the QCL is turned off. From the power spectrum of the demodulated signal, we can validate the reliability of the THz FSO communication system despite the low received optical power of 75 μW of the QCL, which is sufficient for good signal reception. The power spectrum analysis further emphasizes the system's capability to operate reliably at a Gbit/s data rate, with minimal performance degradation from system noise and propagation losses.

The eye diagram in Fig. 4d serves as a tool for assessing transmission quality, where successful communication is indicated by an open eye[55]. A total of $10^7$ symbols were used to plot the eye diagram over a 2 ms time span

of the received signal without any filtering process. It can provide critical insights into system performance, including timing jitter, inter-symbol interference, and signal quality. The eye diagram is plotted across the bit interval, capturing the temporal characteristics of the transmitted data stream and received signal amplitude, indicating the levels of the "0" and "1" bits. The color scale represents the signal trace density, with warmer colors (red/yellow) corresponding to higher counts and cooler colors (blue) denoting lower-count regions. The diagram exhibits a clear eye with well-defined transitions between "0" and "1" bits, indicating minimal distortion and inter-symbol interference. The eye pattern demonstrates the system's ability to maintain timing precision and amplitude integrity over the free-

Fig. 4 | **THz FSO communication with 1 Gbit/s data rate signal transmission. a** The transmitted NRZ-OOK message signal (blue trace) at 1 Gbit/s, showing clean binary transitions generated by the AWG. **b** The demodulated signal (red trace) received after 0.5 m of free-space transmission, demonstrating successful recovery of the transmitted waveform with minimal distortion. **c** Power spectrum of the demodulated signal (red trace) and the noise floor (gray trace) when the QCL is turned off. **d** The eye diagram is plotted across the bit interval, capturing the temporal characteristics of the transmitted data stream and received signal amplitude, indicating the levels of the "0" and "1" bits. The color bar represents the hit count of the signal at each point in the eye. The eye diagram shows a clear pattern indicative of high signal integrity, validating the reliability of the THz FSO communication system for high-speed signal transmission.

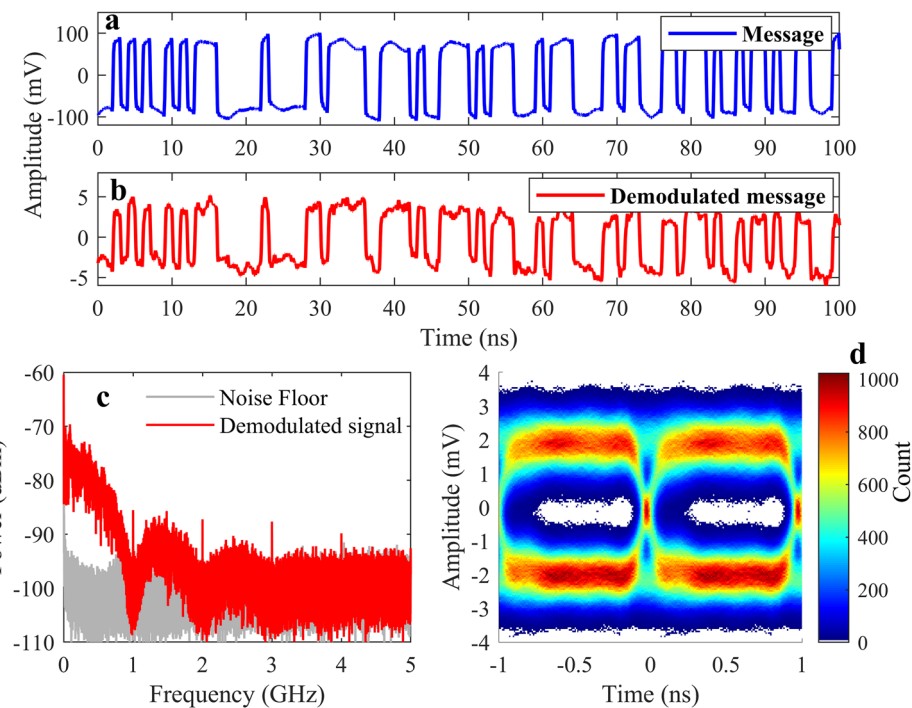

space transmission path. This confirms the robustness of the link and the compatibility of the THz QCL with NRZ-OOK modulation, indicating that it can effectively support communication at multi-gigabit-per-second speeds.

## Transmission performance analysis of the THz FSO communication system

We verify the transmission performance of the THz FSO communication system by evaluating the BER of the demodulated signal. The BER is calculated using the direct error-counting method described in the "Methods" section. Through quantitative analysis of BER, we examine the performance of the THz FSO communication system operating at data rates of 1, 2, 3, and 4 Gbit/s using an NRZ-OOK transmitted signal format. The received optical power is varied using a pair of crossed polarizers to attenuate the signal, facilitating a thorough assessment of the system sensitivity. The received optical power is determined by relating the QCL absolute power to the DC voltage at the receiver bias-tee's DC voltage port.

A PRBS of length $2^{15}-1$ is sent over the channel, and the BER is calculated by comparing the sent and received messages (see Methods) as a function of the received optical power for each data rate. The BER results are summarized in Fig. 5, where data rates of 1, 2, 3, and 4 Gbit/s are compared. A total of $1.98 \times 10^6$ received bits was tested to evaluate the BER over a 2 ms time span. At the lowest data rate of 1 Gbit/s (black circles), the system achieves a BER below $3.8 \times ^{-3}$ at a received power of approximately 30 μW. This corresponds to the BER threshold for hard-decision forward error correction (HD-FEC) with a 7% overhead[56]. Here, 7% overhead indicates that for every 100 bits of original data, an additional 7 bits of redundant data are added, resulting in a total of 107 bits transmitted. This redundancy allows the receiver to correct errors up to a certain BER threshold. The HD-FEC decoding scheme with a 7% overhead typically specifies a BER threshold at approximately $3.8 \times 10^{-3}$ (3.8 errors per 1000 bits). If the received BER is at or below this threshold, the FEC scheme can reliably correct errors, ensuring virtually error-free data reception after correction.

In general, our measurements align with the expectation that BER improves with received power for all data rates (the BER curves slope downward). In an AWGN system, the theoretical expectation is that an increase of 3 dB THz received optical power, leading to a 4-times increase in

electrical signal power from our direct detection system, is expected for a four times change in the data rate while maintaining the same BER in an OOK system. However, our observed increase from 1 to 4 Gbit/s (+1.2 dB per four times increase) is somewhat more modest, likely because other factors, such as a slightly suboptimal threshold determination and the measured noise power spectral density of the detector reducing as the frequency is increased (Fig. S3 of Supplementary Note 2). Nonetheless, the direction and magnitude of the required power increase demonstrate that higher rates demand higher SNR. We also note when the received optical power is increased from 10 to 20 μW, less than one order of magnitude improvement, and from 20 to 40 μW about two orders of magnitude in BER improvement, reaching to error-free performance at higher powers, with no error floors observed down to a BER of ~$10^{-5}$. For maximum received power, we estimate an SNR of 13.5 dB for the demodulated signal (see Table S1 in Supplementary Note 4 for details).

The results show that the THz QCL-based communication system performs reliably over the 0.5 m free-space link for all tested data rates, with increasing power requirements for higher data rate NRZ-OOK signals. The observed trade-off between data rate and optical power highlights the importance of optimizing the QCL output power and detector sensitivity for practical implementation. The time series, spectra, and eye diagrams of the demodulated OOK signal for 2, 3, and 4 Gbit/s are shown in Fig. S6 in the Supplementary Note 5. Furthermore, Fig. 5 shows that the BER reaches below $1 \times 10^{-4}$ for all the data rates, indicating the THz FSO communication system exhibits an excellent transmission performance at least up to 4 Gbit/s given the QCL optical power of 75 μW.

## Transmission performance of THz FSO system for PAM-4 signal

We now examine the transmission capabilities of the THz FSO communication system using a PAM-4 signal. This multi-level modulation scheme encodes two bits per symbol across four amplitude levels, effectively doubling the data rate for a given symbol rate compared to binary formats like NRZ-OOK. The system operation parameters, such as QCL device drive current, temperature, and transmitter/receiver operating condition, are kept the same as those for the NRZ-OOK signal. We consider a data rate of 2 Gbit/s (1 Gbaud) for PAM-4 in the study. The transmitted PAM-4 signal, as shown in Fig. 6a, illustrates the four distinct amplitude levels corresponding to binary values {00, 01, 10, 11}.

**Fig. 5 | THz FSO communication system performance through BER analysis.** BER as a function of received optical power for NRZ-OOK signals transmitted at 1 Gbit/s (black circles), 2 Gbit/s (red squares), 3 Gbit/s (blue diamonds), and 4 Gbit/s (green triangles) using a THz QCL over a 0.5 m free-space link. The BER decreases with increasing received optical power for all data rates, demonstrating improved signal quality. The system achieves BERs below the HD-FEC threshold (black dashed line) of $3.8 \times 10^{-3}$ at received powers of approximately 30 μW for 1 Gbit/s, 34 μW for 2 Gbit/s and 3 Gbit/s, and 40 μW for 4 Gbit/s. These results highlight the trade-off between data rate and the required optical power to reach the HD-FEC threshold.

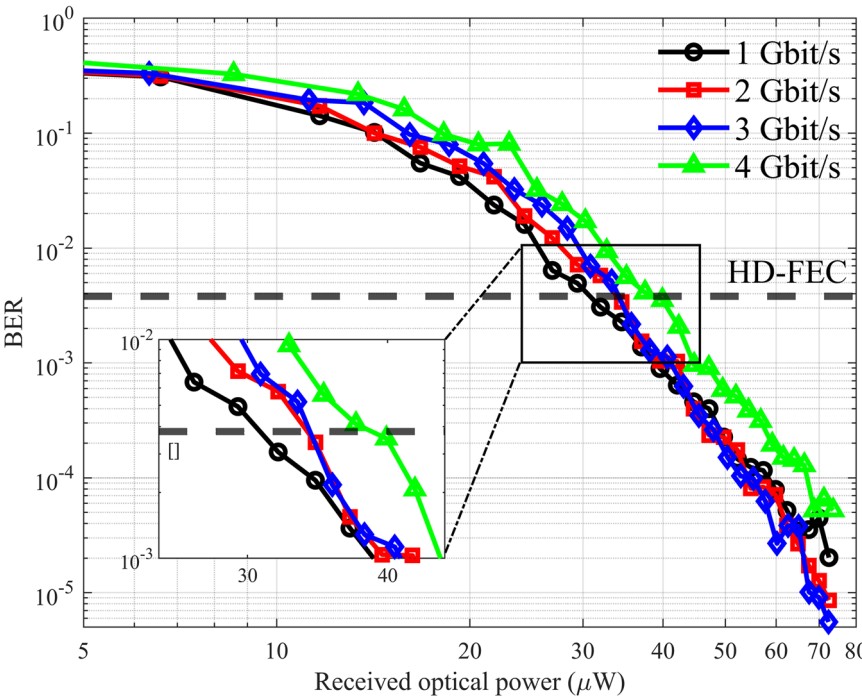

Figure 6 b displays the demodulated signal after 0.5 m of free-space transmission. The PAM-4 signal is more strongly affected by noise than the NRZ-OOK signal, owing to the narrower amplitude separation between the symbols. As such, there is an increased SNR requirement to achieve the HD-FEC threshold for BER. This SNR requirement effect is most clearly seen in the eye diagram of the received PAM-4 signal, Fig. 6d.

Unlike NRZ-OOK, where a single eye opening distinguishes between two levels, PAM-4 should exhibit three distinct eye openings between the four amplitude levels. However, compared to OOK, the eye diagram here demonstrates noticeable distortion, particularly in the lower part of the eye. In our measured PAM-4 eye (Fig. 6d), the eye openings are partially closed and uneven, indicating significant noise and some level of distortion. The lowest and highest levels (corresponding to symbols "00" vs "11") are furthest apart and most distinguishable, while the intermediate levels ("01" and "10") have smaller separations and are more prone to overlap due to noise. This degradation highlights a key challenge of using PAM-4 in THz FSO communication, as the reduced amplitude spacing between levels makes the signal more vulnerable to noise. At 2 Gbit/s (1 Gbaud) PAM-4, the eye diagram (Fig. 6d) should show three eye openings corresponding to the four signal levels. Compared to NRZ-OOK, where only one eye opening is required, this suggests that approximately three times more THz optical power is required to achieve the same BER. The NRZ-OOK at a data rate of 1 Gbit/s (1 Gbaud) requires 30 μW of optical power to reach the HD-FEC BER threshold. From this, we estimate that ~90 μW would be required to bring the BER down to $3.8 \times 10^{-3}$ for 2 Gbit/s PAM-4. Consequently, the BER for PAM-4 at 75 μW of received power exceeds the HD-FEC threshold, indicating insufficient power for error-free communication. The poorer performance is attributed to the 4-level format's higher SNR requirement and the nonlinearity in the QCL modulation, which compresses the PAM-4 levels. Despite this, an attempt to recover a 2 Gbit/s PAM-4 signal (Fig. 6b) demonstrates the feasibility of advanced multi-level modulation formats with a THz QCL, albeit with the need for further system improvements.

The power spectral density of the demodulated signal (red trace), shown in Fig. 6c, provides further insights into the PAM-4 transmission characteristics in the frequency domain. The minima in the spectrum at 1, 2, 3, and 4 GHz are indicative of PAM-4 with a 2 Gbit/s (1 Gbaud) data rate. A PAM-4 modulated signal transmits twice the information per symbol as NRZ-OOK. As such, a 2 Gbit/s PAM-4 scheme requires a symbol rate of

1 Gbaud, but this is at the expense of higher signal complexity and, hence, susceptibility to amplitude noise. Any bandwidth limitations in the system can lead to filtering effects that attenuate higher-frequency components. This spectral roll-off could degrade the amplitude of the demodulated signal.

## Discussion

In the previous sections, we established a multi-gigabit-per-second FSO communication system using a THz QCL source with a 2.4 THz dominant mode and a room-temperature Schottky barrier diode receiver. We demonstrated that our system supports data transmission with 7% overhead HD-FEC for an NRZ-OOK signal, and we evaluated its performance under varying optical power levels and data rates. While the optical power was modest, it is adequate for robust data transmission over the 0.5 m free-space link, allowing us to achieve a BER below the HD-FEC threshold ($3.8 \times 10^{-3}$) at a received optical power of approximately 30 μW for 1 Gbit/s data rate. Furthermore, the BER reaches below $1 \times 10^{-4}$ for data rates tested up to 4 Gbit/s signals, confirming multi-gigabit-per-second communication speeds. The NRZ-OOK modulation scheme used here is well-suited to the noise characteristics of the THz FSO communication system, and the robustness and simplicity of the encoding enable high-speed, high-quality data transmission.

We have tested our communication system with PAM-4 signal transmission. In contrast to NRZ-OOK, the PAM-4 modulation format is more sensitive to noise and to the linearity of the QCL power modulation, owing to the reduced symbol amplitude separation. The results indicate that the PAM-4 signal at 2 Gbit/s requires approximately 90 μW of received optical power to achieve the HD-FEC threshold. Our results show that while PAM-4 offers higher spectral efficiency than OOK, it also introduces greater sensitivity to noise and amplitude distortion. Successful PAM-4 data transmission could be achieved with a QCL by increasing the modulated power (from 75 to 90 μW in this case), increasing receiver sensitivity, and implementing digital signal processing techniques, such as equalization, to mitigate distortions. Higher power QCL sources could be achieved by the use of larger laser ridges, optimized waveguide extraction efficiency, or more efficient active region designs.

To significantly enhance THz link performance, improvements are required on both ends: higher-power, broadband THz QCLs and more sensitive, low-noise THz receivers. Our current link is limited by both the

**Fig. 6 | THz FSO communication system with PAM-4 signal transmission. a** Transmitted PAM-4 message signal (blue trace) at a data rate of 2 Gbit/s, showing four distinct amplitude levels corresponding to binary values {00, 01, 10, 11}. **b** Demodulated PAM-4 signal (red trace) after 0.5 m of free-space transmission, demonstrating signal recovery with distortion. **c** Power spectral density of the received signal (red trace). **d** Eye diagram of the demodulated PAM-4 signal. The color bar represents the hit count of the signal at each point in the eye.

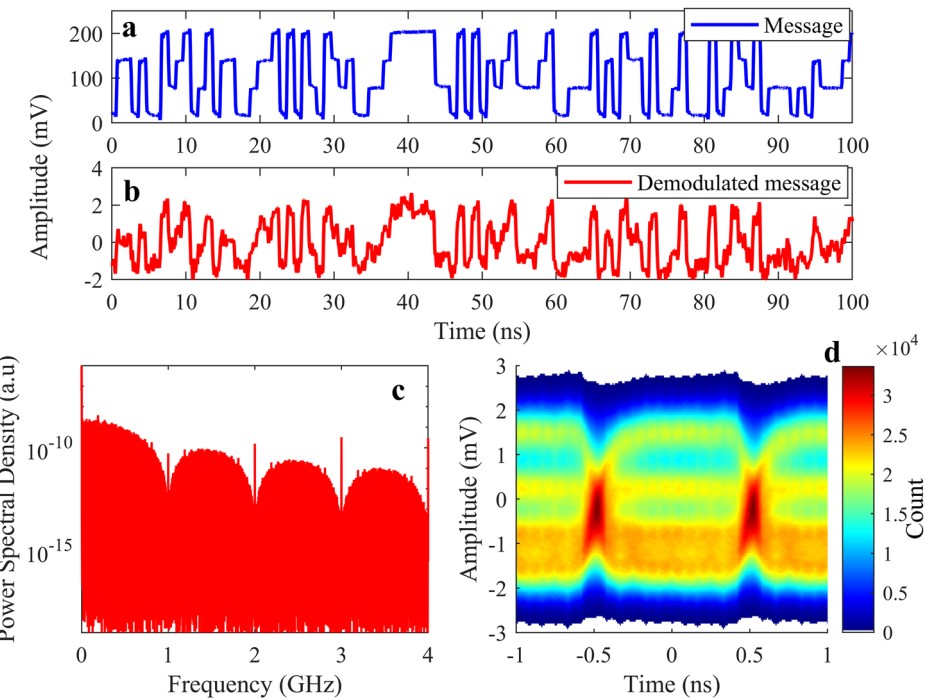

source and the detector. The THz QCL offers a limited THz output power (~75 μW), which is enough for a 0.5 m link at 4 Gb/s OOK. Higher-order modulation, such as PAM-4, clearly shows the transmitter power as a limiting factor, since the available power could not maintain a low BER. The QCL's modulation bandwidth is approximately 5 GHz in our setup, which also limits the achievable symbol rates. On the other hand, the Schottky diode receiver is noise-limited: its room-temperature NEP (~35 pW/Hz$^{0.5}$) requires tens of μW of received power for BER around ~$3.8 \times 10^{-3}$. In our demonstration, the source was the main limitation; however, future THz links will need improvements in both the source and detector, such as higher CW output QCLs in the mW range with multi-GHz bandwidth and receivers with lower NEP or internal gain.

In conclusion, we present an experimental demonstration of a multi-gigabit-per-second FSO communication system utilizing a THz QCL and a room-temperature Schottky barrier diode receiver. This represents a factor of ~200 improvement in data rate compared with any previous THz QCL FSO communication system and is comparable with that of established infrared FSO techniques. We highlight the capability of high-speed FSO communication with reliable performance for both binary and multilevel modulation formats. We demonstrate signal transmission at data rates of up to 4 Gbit/s using NRZ-OOK signals, confirming the potential of THz QCLs in applications such as inter-rack communication in data centers and point-to-point satellite communication. For advanced modulation formats such as PAM-4, our findings suggest that achieving higher data rates will necessitate the development of higher-power QCL sources and/or lower-noise detector technologies. Beyond higher power, future work may explore longer transmission distances along with QCL frequencies carefully matched to atmospheric transmission windows. There is a wideband atmospheric transmission window at 3.4 THz, where many high-performance QCLs have been reported, but the performance of the Schottky barrier detector at that frequency still requires improvement[57]. In regarding linearity considerations, multi-level modulation requires a linear end-to-end response. Our THz link showed some nonlinearity, mainly from the QCL, which reduced the PAM-4 signal quality. In future experiments, this can be addressed by digitally pre-distorting the drive waveform to compensate for the QCL's L-I curve, or by operating the QCL further into its linear region if available. Developing QCLs with a more linear modulation response will directly improve advanced modulation formats, as it allows for more

efficient modulation. Additionally, the integration of FEC techniques could further reduce the optical power requirement, thereby enhancing the robustness of the communication system. Alternative modulation formats, such as OFDM, may provide other routes toward higher efficiency with reduced challenges compared to PAM-4. These results highlight the adaptability and promise of THz QCLs as compact and scalable sources for next-generation wireless networks, opening new opportunities for ultra-fast THz communication technologies.

## Methods
### QCL source
The QCL structure used in this work employed a nine-well GaAs/AlGaAs active region providing gain at frequencies in the range 2.3–2.7 THz. The active region was based on a "hybrid" QCL design, including a resonant-phonon-assisted depopulation stage, modified from ref. 58. A 12 μm thick heterostructure comprising 96 repeat periods was grown by molecular-beam epitaxy on a 76.2 mm semi-insulating GaAs substrate and processed into a metal–metal waveguide structure, using gold–gold wafer bonding[59]. Ridges 70 μm wide were defined by dry etching in a Cl$_2$/Ar plasma. The top contact layer was Ti/Au (10/150 nm thick). Ridges 2 mm long were cleaved and mounted with indium solder on a copper submount together with a coplanar waveguide defined on a printed circuit board. The coplanar waveguide was designed for a 50 Ω impedance and accommodates an SMA connector to facilitate high bandwidth coupling. The QCL device was mounted on the cold finger of a closed-cycle cryostat (Sumitomo CH-204N) and held at a heat sink temperature of 19 K for these measurements.

In Fig. 3b, the QCL's output power (75 μW at 300 mA, 19 K) was measured using a calibrated THz power meter. In our setup, after the two 90° parabolic collimating mirrors, we placed a thermopile-based THz power sensor (Ophir power meter) at the receiver focus to directly measure the emitted CW power. This provided the absolute output power of the QCL in watts, and then this absolute power was calibrated with CW L-I-V.

### Driving electronics
The experimental arrangement is shown in Fig. 2. At the transmitter, NRZ-OOK and PAM-4 digital signals with various data rates were generated offline using Keysight IQtools, as detailed in the next section. The digital samples generated are converted to the analog domain by an AWG (12 GSa/

s, Keysight M8190A). The output of the AWG is adjusted between 280 and 500 mV$_{pp}$, depending on the modulation formats, and is amplified by a low-noise amplifier (ZHL-1042J+) with a bandwidth of 4.2 GHz and a gain of 27 dB. A broadband bias-tee (ZX85-12G-S+) combines the modulation signal with a DC current supplied by a current source (Arroyo Instruments Laser source 4302), with the combined output connected to the QCL bias terminals.

### Receiver system

A pair of 90° off-axis parabolic mirrors, with focal lengths of 50.8 and 152.4 mm, is used to collimate the QCL output beam and focus the THz radiation onto the detector at the receiver. No multi-path signatures, such as delayed replicas, were observed in the received time-domain waveforms or eye diagrams, and the BER versus received optical power (Fig. 5) was monotonic, indicating a single path.

The receiver comprises a WR-0.34 (WM-86) Schottky barrier diode 4th-harmonic mixer (Virginia Diodes Inc.), operated as a direct THz envelope detector in this work. The mixer RF port is coupled to a diagonal feedhorn antenna with a gain of 25 dBi to collect the THz signal, and has 2.1–3.3 THz bandwidth. The IF port of the mixer was connected to a second bias-tee (8812KMF2-26) with a 26 GHz bandwidth, an RF amplifier (ZVA-213UWX-1+) with a 20 GHz bandwidth and a gain of +15 dB. The detector operates on the zero-bias Schottky diode principle, rectifying the THz signal solely through the nonlinearity of its $I$-$V$ curve at zero bias. The THz oscillating field induces a nonlinear current in the Schottky junction, whose time-average produces a DC or low-frequency output proportional to the incident power[60]. This zero-bias operation eliminates shot noise from the bias current, resulting in lower noise than biased detectors. The detection is direct and follows a square-law response at small signals, yielding an output voltage $V_{out} = \mathcal{R}_V \times P_{THz}$ proportional to the incident THz power. Here, $\mathcal{R}_V$ is the voltage responsivity, measured to be ~31 V/W. We determined the receiver system noise-equivalent power (NEP) to be approximately 35 pW/Hz$^{0.5}$. Further details on the responsivity and noise characteristics of the SBD are given in Supplementary Note 2. A key advantage of Schottky diode detectors is their ultra-fast response. The WR-0.34 Schottky device is intrinsically capable of detecting modulation up to tens of GHz. A similar WR-0.34 harmonic mixer has been tested with IF outputs up to 40 GHz[44]. In our setup, the IF chain utilized a 26.5 GHz bias-tee and a 20 GHz amplifier. However, the oscilloscope limited the analyzed bandwidth to 4 GHz. Consequently, the measured end-to-end system bandwidth is ~5 GHz (Fig. 3c).

The received optical power is obtained by calibrating the Schottky barrier diode's DC output against a measured absolute power QCL. To determine received optical power, we monitored the DC voltage from the Schottky diode (via the bias-tee's DC port), which is proportional to incident THz power. We calibrated this DC voltage against the known 75 μW output. Once calibrated, we used the SBD's DC voltage as an indicator of received power when we introduced additional attenuation. We varied the received power using crossed polarizers. For each polarizer angle, we recorded the DC level from the SBD. Using the initial calibration (voltage-to-power mapping), we calculated the received optical power corresponding to that DC level. The THz power envelope on the IF port was recorded on a real-time digital storage oscilloscope (Keysight DSOS404A) with a 4-GHz bandwidth and 10-bit ADC capable of 20 GSa/s. The oscilloscope captured the received signal, and the converted digital samples were processed offline to estimate the BER, as detailed in the next section.

### Offline digital signal processing and BER estimation

At the transmitter, NRZ-OOK/PAM-4 digital signals were generated offline using Keysight IQtools. A PRBS-15 sequence (length $2^{15}-1$) is utilized for both NRZ-OOK and PAM-4 format signals. A sample rate of 12 GSa/s is set to match the sampling rate of the AWG. The symbol rate, measured in Gbaud, was adjusted to achieve various data (bit) rates for the modulated signal. To find the bit rate (Gbit/s), Gbit/s = bits/symbol × Gbaud was used. For a binary NRZ-OOK signal, the bit rate equals the baud rate. For

example, a 1 Gbaud NRZ-OOK signal corresponds to a bit rate of 1 Gbit/s. In $M$-level PAM signals, there are $\log_2(M)$ bits per symbol. Therefore, for PAM-4, each symbol represents 2 bits, leading to a bit rate of Gbit/s = 2 × Gbaud.

BER is estimated through offline signal processing of the demodulated signals for both NRZ-OOK and PAM-4 formats. The oscilloscope captured 2 ms of waveform per measurement, yielding 2–8 million bits depending on data rate. We first filter the demodulated signal using a fifth-order Bessel filter with a cut-off frequency of 0.7 × symbol rate of the data. The symbol rate is the same as the data rate for NRZ-OOK and half of the data rate for the PAM-4 signal. Synchronizing the AWG and oscilloscope with a 10 MHz reference clock ensures that the sampling time is consistent across the data. A PRBS of length $2^{15}-1$ is generated to align with the bit sequence sent by the AWG. Then, we sampled the data to obtain the received bit sequence using a sampling time and a threshold value estimated from the eye diagram for the first 1000 bits. We synchronize the sampled data with the PRBS by calculating the BER for every possible alignment (direct error counting); a minimum BER indicates alignment. Now, with the synchronization described above, we perform a sweep of the sampling time and threshold value for all samples within a bit period, using a fixed value of equally spaced threshold values ranging from −20% to +20% of the signal amplitude around the signal mean. The sampling time and threshold value that produced the lowest overall BER were considered to be the optimum BER values. The reported BER was the minimum across the entire period of the captured data for a fixed optimum sampling time and threshold value.

## Data availability

The data associated with this paper are publicly available from the University of Leeds Data Repository at https://doi.org/10.5518/1754(ref. 61).

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

## Acknowledgements

This work was supported by UK Research and Innovation through the Programme Grant TeraCom (EP/W028921/1) and Future Leader Fellowship (MR/S016929/1 and MR/Y011775/1). We would like to acknowledge Dr Purushothaman Narasimhan and Prof. Kevin Morris from the School of Electronic and Electrical Engineering, University of Leeds, for providing the arbitrary waveform generator for this experiment.

## Author contributions

J.E. and J.R.F. developed the experimental setup and performed the measurements. J.E. analyzed the data with support from M.F. and J.R.F. The QCL structure was grown by L.L. under the supervision of E.H.L. The device was processed by M.S. under the supervision of E.H.L. and A.G.D. A.G.D., A.S., E.H.L., J.R.F., A.V., and M.F. supervised the project. The paper was written by J.E., J.R.F., and M.F. with contributions from all authors.

## Competing interests

The authors declare no competing interests.
