## [Transparent Peer Review file · Communications Physics]

Free-space Optical Communications at 4 Gbit/s Data Rate with a Terahertz Laser

Corresponding Author: Dr Jayaprasath Elumalai

Version 0:

Reviewer comments:

Reviewer #1

(Remarks to the Author)

This manuscript reports on multi-gigabit-per-second data transmission at 2.4 THz, using a directly modulated THz cascade laser (QCL) and a Schottky barrier diode (SBD) receiver. The authors demonstrate maximum data rates of 4 Gb/s using NRZ-OOK and 2 Gb/s using PAM4 modulation. Compared with previously reported results, this work sets a clear milestone by surpassing prior data rate records by over two orders of magnitude. In my view, this is a highly impressive achievement that offers promising potential to help bridge the long-standing "THz gap."

However, I have several comments and questions that I believe the authors should address before the manuscript can be recommended for publication. These suggestions are made with the aim of maintaining scientific rigor.

1. The THz QCL in this work operates at 10 K. While the authors mention in the Introduction that some THz QCLs can operate with Peltier cooling, the reliance on a cryocooler remains a significant limitation for practical applications. Please clarify whether Peltier-cooled THz QCLs are currently capable of supporting data transmission at these rates. If not, please elaborate on the current limitations and suggest potential pathways toward such operation.
2. Is the QCL used in this study a DFB laser or a Fabry–Pérot cavity? Given the presence of multiple emission modes, I would assume it is a Fabry–Pérot structure. Please confirm and clarify.
3. In my opinion, the key enabler of this work, besides the THz QCL, is the SBD receiver. Although this device has been reported previously (e.g., in Ref. [44]), it deserves a more detailed description here, given its key role in the communication system. Please describe its structural configuration, operating principle, and provide its noise characteristics and bandwidth performance. This information is essential for readers to fully understand how such high-speed transmission was achieved.
4. Building on the previous point, I strongly encourage the authors to include a comprehensive system-level noise/SNR and power budget. Given that the employed THz QCL only emits $\sim 80 \mu\text{W}$ power, much lower output than the mid-infrared QCLs, the system's ability to achieve such performance likely relies on the exceptional characteristics of the SBD detector. A quantitative budget would greatly enhance the technical stringency of the paper.

Minor comments:

5. In Figure 2, the gain of the Tx RF amplifier is listed as 27 dB in both the main text and figure caption, but is shown as 25 dB in the figure itself. Please correct the discrepancy.

6. Also in the caption of Figure 2, the term "demodulated signal" should be replaced with "received signal," since the signal before the bias-tee has not yet been demodulated.

Reviewer #2

(Remarks to the Author)

This paper presents signal generation at a 2.4 THz carrier frequency using a QCL, and demonstrates 4 Gb/s NRZ signal

transmission over a 0.5 m wireless link. The results are impressive; however, there are several questions (especially on the of the transmission results) that need to be addressed.

1. In the paper, the communication at a 2.4 THz carrier frequency is described as optical communication. Although the choice of terminology is up to the authors, in my opinion, "optically generated THz wave" or "optically generated sub-mm wave" is more appropriate. Aside from the terminology, the expressions "THz" and "optical" are used interchangeably in the paper.
2. Please include details on how the THz power was measured in Figs. 3b and 5.
3. In Fig. 3b, can you comment on why the THz power does not increase linearly with the QCL current? I wonder whether this is due to a measurement error or a characteristic of the QCL.
4. If possible, please describe the x-axis in Fig. 5 using a dB scale for better readability.
5. In the third paragraph of the Results section, the paper states: "we assume that the RIN is negligible compared to the electrical noise produced by the amplifiers." However, in most cases, the effect of the noise produced by the amplifiers is negligible compared to that of the thermal noise in the Schottky diode. Please verify this statement.
6. In Fig. 5, can you comment on why the BER curves are similar for 1, 2, and 3 Gb/s? Generally, if the data rate is doubled, the required received power should also be doubled to achieve the same SNR/BER.
7. In the PAM-4 modulation experiment, did you optimize the bias current and modulation power? Based on the eye diagram in Fig. 6d, it appears that the modulation conditions may not have been fully optimized. Additionally, in Fig. 6a, the two messages seem to be unsynchronized. If the current /experiment of the PAM-4 transmission is not sufficiently thorough, it might be worth considering whether this section could be deferred or removed in the current version of the paper.
8. Can you provide any insight into the maximum achievable data rate of the system?

Reviewer #3

(Remarks to the Author)

The article shows results of THz transmission using a QCL, for 4 Gbit/s. The concept of this is not new, even if the data-rate is higher than SoTA for this frequency (carrier) and for the use of QCL. Also, it is difficult to assess the high impact nature of the work related to the main limitations observed in the results.

I have several comments that should be addressed to better assess the potential impact:

- This is not clear what are the main limiting factors in the transmission, the source, the receiver, or a combination of the 2? As the title is emphasizing the THz laser, one could think the receiver is not the limiting part of the system, but it should be proven and if not, authors should explain what are the induced limitations from the receiver used. As simple example, the bandwidth and linearity of the Schottky diode used at receiver should be checked.
- Related to the source, what about the radiation of the QCL signal in the free-space? A proper radiation with no echos is mandatory for telecom applications, was this checked?
- For the BER evaluations, how was this done? When using an ON/OFF modulation, this is common to measure the BER with a bit error rate tester instead of using off-line demodulation. The BER values dependence with sequence length was checked?
- Related to the linearity of the system, for the QCL part, I suggest to authors to analyze the linearity of the modulation of the QCL prior to experiments, to check the linearity level and identify in case the sources of no-linearity.
- Especially, the signal in PAM-4 is (eye pattern) fully closed. Thus, the source
- There is some mistake: authors discussed about "GBaud/s", but this unit doesn't exist: "Baud" means the number of symbols per second, thus doesn't need to add "/s".

Version 1:

Reviewer comments:

Reviewer #1

(Remarks to the Author)

I have carefully gone through the authors' responses and the revisions. I believe the quality of the manuscript has substantially improved. I can recommend accepting the paper for publication as is.

Reviewer #2

(Remarks to the Author)

All of my comments have been appropriately addressed.

Reviewer #3

(Remarks to the Author)

Authors did corrections, but some points remains unclear in my perspective.

For PAM4, I don't see really the interest here as the signal quality is very low and the eye pattern closed. It would be worth to determine the required power margin to make the system running, and discuss how it could be achieved (or not) using the QCL approach?

About BER curves, something is surprising: when increase the data-rate by a factor of 4, the required power level remains more or less constant. This is not expected from physics.

Authors said that they expect 3 dB extra power when multiply the data-rate by 4. This is somehow strange as when increasing the data-rate by 4, the spectrum frequency extension is also increased by 4. Thus integrated noise bandwidth is increased by 4, which is 6 dB more, and to compensate this, an extra 6dB power is required. Can authors discussed their 3 dB prediction?

End of page 6, there is a mistake : « ... a low-noise amplifier with +15dB gain to improve the signal-to-noise ratio”. This is not correct: any amplifier degrades the SNR. The LNA increases the signal to make it sufficiently powerfull for the receiver, but degrades the SNR of the signal in any case.

Version 2:

Reviewer comments:

Reviewer #3

(Remarks to the Author)

Authors improved the manuscript, it is now suitable in my perspective.

Point-by-point responses to the reviewers' comments

Dear Reviewers,

Thank you for your comments concerning our manuscript entitled "Free-space Optical Communications at 4 Gbit/s Data Rate with a Terahertz Laser" (Manuscript ID: COMMSPHYS-25-0640-T). We have carefully reviewed the comments and made the relevant modifications, which are highlighted in blue in the revised manuscript. The point-by-point responses to the reviewer's comments and modifications to the manuscript are listed below.

Reviewer #1 (Remarks to the Author):

This manuscript reports on multi-gigabit-per-second data transmission at 2.4 THz, using a directly modulated THz quantum cascade laser (QCL) and a Schottky barrier diode (SBD) receiver. The authors demonstrate maximum data rates of 4 Gb/s using NRZ-OOK and 2 Gb/s using PAM4 modulation. Compared with previously reported results, this work sets a clear milestone by surpassing prior data rate records by over two orders of magnitude. In my view, this is a highly impressive achievement that offers promising potential to help bridge the long-standing "THz gap."

However, I have several comments and questions that I believe the authors should address before the manuscript can be recommended for publication. These suggestions are made with the aim of maintaining scientific rigor.

Response: We thank the reviewer for the thoughtful evaluation of our work. We are grateful that the reviewer found the results compelling and the step toward bridging the "THz gap" noteworthy.

Comment 1. The THz QCL in this work operates at 10 K. While the authors mention in the Introduction that some THz QCLs can operate with Peltier cooling, the reliance on a cryocooler remains a significant limitation for practical applications. Please clarify whether Peltier-cooled THz QCLs are currently capable of supporting data transmission at these rates. If not, please elaborate on the current limitations and suggest potential pathways toward such operation.

Response: We agree that the current requirement for cryogenic cooling is a practical limitation of THz QCLs. In our experiment, the QCL was operated in a closed-cycle cryostat at a base temperature of ~20 K (in continuous-wave mode). While recent advances in QCL design have enabled Peltier-cooler operation, these high-temperature demonstrations require pulsed operation and very high drive currents [DOI: 10.1364/OE.27.020688]. In continuous-wave (CW) mode, the highest reported operating temperature for a THz QCL with measurable output power is ~130 K [DOI: 10.1109/TTHZ.2019.2935337], well below room temperature. Thus, at present, a 2.4 THz QCL still necessitates cryogenic cooling for CW operation. We also note that achieving CW operation in compact

coolers, including Peltier and Stirling coolers, is an active topic of research. We have revised the **Results: THz FSO communication link section (page 4, paragraph 1)** to state that our QCL requires cryogenic cooling.

Comment 2. Is the QCL used in this study a DFB laser or a Fabry–Pérot cavity? Given the presence of multiple emission modes, I would assume it is a Fabry–Pérot structure. Please confirm and clarify.

Response: We apologize for not explicitly stating the laser cavity type in the original manuscript. The 2.4 THz QCL is a Fabry–Pérot cavity QCL with a metal-metal waveguide (dimensions 2 mm × 70 μm). Although it operates on multiple longitudinal modes, there is one dominant mode at ~2.4 THz. We have now explicitly described the laser as a Fabry–Pérot QCL and noted the multi-mode emission (the primary mode at 2.4 THz is ~15 dB stronger than the nearest side mode at ~2.63 THz) in the **Results: THz FSO communication link section (page 4, paragraph 1)**.

Comment 3. In my opinion, the key enabler of this work, besides the THz QCL, is the SBD receiver. Although this device has been reported previously (e.g., in Ref. [44]), it deserves a more detailed description here, given its key role in the communication system. Please describe its structural configuration, operating principle, and provide its noise characteristics and bandwidth performance. This information is essential for readers to fully understand how such high-speed transmission was achieved.

Response: We agree with the reviewer that the Schottky-barrier diode receiver is a critical part of the system. Since it is a commercial product, the details of its internal structure are not publicly available. We have now expanded the description to outline the available information from the manufacturer clearly. The receiver utilizes a Virginia Diodes Inc. (VDI) WR0.34 Schottky barrier diode, which can function as a 4th-harmonic mixer, operated here as a direct (envelope) detector. The device consists of a zero-bias Schottky diode integrated with a diagonal feedhorn antenna (aperture ~0.38 mm) in a waveguide block. No local oscillator was used; the diode simply rectifies the incoming THz field. We have added these details in the designated section of the revised manuscript in the **Methods: Receiver system (page 14, paragraph 2)**, noting that the diode was used at zero bias and coupled via a 90° off-axis parabolic mirror into its feedhorn. The detector's IF port was connected to a bias-tee and low-noise amplifier as described. **Fig. 2** has been updated to label the diode as a “Schottky diode harmonic mixer (zero-bias, acting as THz envelope detector)” for clarity.

We have also briefly described SBD's operating principle and bandwidth performance in the revised manuscript in the **Methods: Receiver system section (page 14, paragraph 2)**. We have also add a section in the **Supplementary Information (section 2, page 3, paragraph 1)** describing the responsivity and noise characteristics of the detector.

Comment 4. Building on the previous point, I strongly encourage the authors to include a comprehensive system-level noise/SNR and power budget analysis. Given that the employed THz QCL only emits $\sim 80 \mu\text{W}$ power, much lower output than the mid-infrared QCLs, the system's ability to achieve such performance likely relies on the exceptional characteristics of the SBD detector. A quantitative budget would greatly enhance the technical stringency of the paper.

Response: We appreciate the reviewer's comment on the power budget analysis of the system. We now detail the link budget and signal-to-noise ratio (SNR) for the 2.4 THz free-space link, using the measured detector parameters and known transmitter characteristics. The link configuration consists of a 2.4 THz QCL transmitter and a Schottky diode detector as the receiver, separated by 0.5 m of free space. At the receiver focus, we measured P_{THz} approximately $75 \mu\text{W}$ at a drive current of 300 mA and a temperature of 19 K. With a modulation of 19dBm applied, we estimate a modulated Power of $50 \mu\text{W}$. For BER sweeps, we introduced calibrated attenuation and used the Schottky diode DC output to obtain received power [**Methods: Receiver system section (page 14, paragraph 3)**]. The receiver system responsivity was calculated to be $\sim 176 \text{ V/W}$. This was then combined with the integrated noise PSD measured after pre-amplification to obtain a noise equivalent power (NEP) for the receiver system of $35 \text{ pW/Hz}^{0.5}$. The power budget is summarised in Table 1.

Table 1: System-level noise/SNR and power budget analysis

Parameter	
Modulation Power (P_{mod})	50 μW
Bandwidth (B)	4 GHz
Receiver System Responsivity	176 V/W
System NEP	35 $\text{pW/Hz}^{0.5}$
SNR [$P_{mod}/(\text{NEP} \times \sqrt{B})$]	13.6 dB

The end-to-end system 3 dB bandwidth $\sim 5 \text{ GHz}$ (measured) sets the modulation bandwidth limitation, while the NEP sets the power margin. Overall, the link budget analysis indicates that the system operated with a comfortable SNR margin at 4 Gbit/s over 0.5 m. Future improvements (higher QCL power or a more sensitive receiver) could allow longer ranges or higher data rates. We have now incorporated this explanation in the revised manuscript of the **Discussion section (page 12, paragraph 3)** and added a new section in **Supplementary Information (page 6, section 4, paragraph 1)**.

Minor comments:

Comment 5. In Figure 2, the gain of the Tx RF amplifier is listed as 27 dB in both the main text and figure caption, but is shown as 25 dB in the figure itself. Please correct the discrepancy .

Response: We appreciate the reviewer's attention to this error. This has now been corrected in **Fig. 2**.

Comment 6. Also in the caption of Figure 2, the term "demodulated signal" should be replaced with "received signal," since the signal before the bias-tee has not yet been demodulated.

Response: We appreciate the reviewer's concern. As the Schottky barrier diode acts as an envelope detector in this work, the THz carrier is inherently rejected. We have, though, replaced the description in the caption of **Fig. 2**, in line with the reviewer's suggestion.

Reviewer #2 (Remarks to the Author):

This paper presents signal generation at a 2.4 THz carrier frequency using a QCL, and demonstrates 4 Gb/s NRZ signal transmission over a 0.5 m wireless link. The results are impressive; however, there are several questions (especially on the analysis of the transmission results) that need to be addressed.

Response: We are grateful to the reviewer for the positive assessment of the work and for highlighting the need to sharpen the analysis. We have made focused revisions to address the points raised in the revised manuscript.

Comment 1. In the paper, the communication at a 2.4 THz carrier frequency is described as optical communication. Although the choice of terminology is up to the authors, in my opinion, "optically generated THz wave " or "optically generated sub-mm wave" is more appropriate. Aside from the terminology, the expressions "THz" and "optical" are used interchangeably in the paper.

Response: We understand the reviewer's concern and have taken steps to clarify our usage of the term "optical." In this work, the transmitter is a laser (a THz QCL), so we use "free-space optical (FSO)" in the laser-based sense of optical wireless links. We used "free-space optical (FSO) communication" in a broad sense, encompassing unguided wireless links that use electromagnetic carriers beyond the radio/microwave range (historically, this includes infrared and visible light). Terahertz waves (1-10 THz) border the far-infrared region, so it is indeed a grey area in terminology.

In the **Introduction section (page 2, paragraph 1)**, when first mentioning free-space optical links, we now explicitly state: "*The 'optical links' refer to free-space laser-based links, extended here to the terahertz frequency band*". We justify this terminology by noting that the THz link shares many characteristics with conventional optical wireless links (high directionality, use of laser sources, line-of-sight propagation), and that our work extends the concept of optical wireless communications into the terahertz band. By making these clarifications, we ensure readers will not mistake our meaning. We

also updated **keywords** to include “terahertz communications” alongside “optical wireless communications” to guide readers.

Comment 2. Please include details on how the THz power was measured in Figs. 3b and 5.

Response: We appreciate the reviewer's concern. We have now provided a detailed explanation of our THz power measurement methods. The QCL's output power ($\sim 75 \mu\text{W}$ at 300 mA, 19 K) was measured using a calibrated terahertz power meter. In our setup, after a pair of 90° off-axis parabolic collimating mirrors, we placed a thermopile-based THz power sensor (Ophir power meter) at the receiver focus to measure the emitted CW power directly. This provided the absolute output power of the QCL in watts, and then this absolute power was calibrated with CW L-I-V. We mention this calibration in the revised **Methods: QCL source section (page 13, paragraph 2)**.

To measure the received optical power, we employed an indirect yet accurate approach: we correlated the Schottky diode's DC output voltage (from the bias-tee DC port) with the incident THz power. When the QCL is emitting, the zero-bias diode produces a small DC rectified voltage proportional to the incident power. We first calibrated this by comparing the DC voltage at full QCL bias to the absolute power reading of the THz power meter. Once calibrated, we used the diode's DC voltage as an indicator of received power when we introduced additional attenuation. Specifically, as noted in the manuscript, we varied the received power using crossed polarizers - for each polarizer angle, we recorded the DC level from the diode. Using the initial calibration (voltage-to-power mapping), we calculated the received power corresponding to that DC level. This is how we obtained the “received optical power” values on the x-axis of Fig. 5. We have clarified this procedure in the revised manuscript in the **Methods: Receiver section (page 14, paragraph 3)**.

Comment 3. In Fig. 3b, can you comment on why the THz power does not increase linearly with the QCL current? I wonder whether this is due to a measurement error or a characteristic of the QCL .

Response: We appreciate reviewers' comments on the nonlinearity of the QCL power. The nonlinearity of the QCL's L-I curve is a well-known characteristic of quantum cascade lasers and is primarily due to thermal effects in the laser active region. We have expanded our discussion of the QCL's CW L-I-V characteristics (referencing Fig. 3b in the manuscript) to explain this. Our device is a 2.4-THz double-metal QCL with $I_{\text{th}} = 175$. Above the threshold, the power increases, but in CW operation, the active region self-heats as the V-I curve rises. The resulting temperature rise (i) shifts the gain peak and (ii) increases internal loss, so the differential efficiency decreases, and the L-I curve becomes sub-linear. The minor kinks/steps in the red trace occur when the longitudinal Fabry-Pérot modes shift relative to the gain peak and different modes cross their respective thresholds. As the current approaches ~ 360 mA, the device exhibits transport domain formation and further self-heating, which reduces the effective gain length and produces the observed power peak and roll-over. Thermal roll-over [DOI:10.1063/1.1886266], mode-dependent steps [DOI:10.1063/1.4949528], and roll-over near the

onset of NDR [DOI: 10.1103/PhysRevB.73.033311] characteristics are widely reported for THz QCLs operated in CW. We have clarified and revised the manuscript in the **Results: THz FSO communication link section (page 5, paragraph 1)** and in the **Fig. 3 caption**.

Comment 4. If possible, please describe the x-axis in Fig. 5 using a dB scale for better readability .

Response: We appreciate the reviewer’s suggestion. The range of values on the x-axis is relatively small (-23 dBm to -11 dBm), and as such, we find that readability is poor on a dB scale. We have, though, modified the x-axis to a log scale (with power displayed in linear μW units) in **Fig. 5** in the revised manuscript.

Comment 5. In the third paragraph of the Results section, the paper states: “we assume that the RIN is negligible compared to the electrical noise produced by the amplifiers.” However, in most cases, the effect of the noise produced by the amplifiers is negligible compared to that of the thermal noise in the Schottky diode. Please verify this statement.

Response:

We thank the reviewer for bringing this to our attention. We agree that the dominant source of noise will come from the Schottky diode. We have included more details of the power and noise performance of the receiver system in the supplementary information to quantify the receiver system NEP, responsivity, and present an estimation of the SNR. We have also modified the third paragraph in the **Results section (page 6, paragraph 1)** to account for this and have included estimates of the receiver NEP and system SNR in the **Supplementary Information (page 3, section 2, paragraph 1)**.

Comment 6. In Fig. 5, can you comment on why the BER curves are similar for 1, 2, and 3 Gb/s? Generally, if the data rate is doubled, the required received power should also be doubled to achieve the same SNR/BER.

Response: We appreciate the reviewer’s concern about the BER curves for different data rates. We have now expanded our discussion of Fig. 5 (BER vs. received power for various data rates) to address this question. In the revised text, we point out the following:

- At a given received power, the BER degrades as the data rate increases. This is evident in Fig. 5, where the BER curves shift to the right as the data rates increase. Now to describe this trend in words: “For a fixed received power, higher data rates result in higher BER (lower signal quality), which is expected since a higher-bandwidth signal has increased noise integration and reduced energy per bit.” Essentially, as the bit rate goes up, the receiver’s noise bandwidth increases (approximately proportional to bit rate for NRZ-OOK), so more noise power is collected. This requires a higher received power to maintain the same BER.

- We quantify how much additional power is needed as the data rate increases in our system. The revised manuscript notes: “The system achieves the HD-FEC BER threshold (3.8×10^{-3}) at $\sim 30 \mu\text{W}$ for 1 Gbit/s, $\sim 34 \mu\text{W}$ for 2 Gbit/s and 3 Gbit/s, and $\sim 40 \mu\text{W}$ for 4 Gbit/s.” Thus, going from 1 to 4 Gbit/s required roughly $+10 \mu\text{W}$ more received power to reach the same BER threshold. The theoretical expectation is that an increase of 3 dB THz received optical power is expected for a four times change in the data rate while maintaining the same BER in an OOK system. However, our observed increase from 1 to 4 Gbit/s ($+1.2 \text{ dB}$ per $4\times$ increase) is somewhat more modest, likely because other factors, such as a slightly suboptimal threshold determination or pattern-dependent effects at lower rates, come into play. Nonetheless, the direction and general magnitude of the required power increase align with the idea that higher rates demand higher SNR.
- We highlight the slope of the BER curves: at high BER (left side of each curve), BER drops rapidly with additional power, whereas near the FEC threshold, the curves flatten out. This is a typical behavior; initially, adding a little power dramatically reduces errors (when operating in a low-SNR regime). However, once the BER is already low (near 10^{-3}), further power provides diminishing improvement, as errors are infrequent and dominated by occasional noise spikes. We mention this to illustrate our understanding of the curve’s shape.
- We also added a brief note on the error floors: none of our measured BER curves showed an error floor down to the lowest BER we measured, around 10^{-5} at the highest power for 1 Gbit/s. The BER continued to improve with power, indicating that the system is primarily noise-limited rather than, for example, limited by inter-symbol interference (ISI) or other distortions at those rates. We state that “no floor was observed above 10^{-5} BER, confirming that additive noise is the dominant impairment, and that higher SNR would continue to yield lower error rates.”

In summary, our revised description explicitly connects the BER vs power behaviour to fundamental considerations of SNR and data rate. We have now incorporated this explanation in the revised manuscript of the **Results: Transmission performance analysis of the THz FSO communication system section (page 9, paragraph 2)**.

Comment 7. In the PAM-4 modulation experiment, did you optimize the bias current and modulation power? Based on the eye diagram in Fig. 6d, it appears that the modulation conditions may not have been fully optimized. Additionally, in Fig. 6a, the two messages seem to be unsynchronized. If the current analysis/experiment of the PAM-4 transmission is not sufficiently thorough, it might be worth considering whether this section could be deferred or removed in the current version of the paper.

Response: We acknowledge the reviewer’s concern that our original discussion of the PAM-4 experiment was brief. We have significantly modified the analysis of the PAM-4 results (**Transmission performance of THz FSO system for PAM-4 section**). Key points we now address:

- Eye diagram interpretation: We describe the PAM-4 eye diagram (Fig. 6d) in detail. The revised text explains that, unlike the binary (NRZ-OOK) case, which has a single eye opening, the PAM-4 scheme has three distinct eye openings corresponding to the three decision thresholds between the four amplitude levels. We explicitly note that in our measured PAM-4 eye (Fig. 6d), these eye openings are partially closed and uneven, indicating significant noise and some level of distortion. The lowest and highest levels (corresponding to symbols “00” vs “11”) are furthest apart and most distinguishable, while the intermediate levels (“01” and “10”) have smaller separations and are more prone to overlap due to noise. The reviewer is correct in noting that the “modulation conditions may not have been fully optimized”. Indeed, the level separations could be optimised with pre-processing of the message signal, to compensate for the nonlinearity of the QCL $L-I$ curve. However, the more fundamental issue is the available SNR at this stage.
- Message synchronisation: The reviewer is correct that the segments of the sent and received messages in Fig. 6(a) and (b) were unsynchronized. We have **updated Fig. 6** to show a synchronised plot (i.e., applying the time delay that yields maximal cross-correlation between the two signals). It is important to note, though, that the BER calculation is determined by comparing the demodulated receiver signal with a pure PRBS-15 message, rather than with the AWG output [Fig. 6(a)], so this has no impact on the subsequent analysis or discussion.
- PAM-4 BER performance: Importantly, we clarify how the PAM-4 link performed in terms of BER. In the original submission, we mentioned that more power ($\sim 90 \mu\text{W}$) would be required to reach the FEC threshold for PAM-4, implying that the available $75 \mu\text{W}$ was insufficient to meet the threshold. In the revision, we make this explicit: we state that “at the maximum QCL power ($\sim 75 \mu\text{W}$), the 2 Gbit/s PAM-4 link achieved a BER above the HD-FEC threshold of 3.8×10^{-3} .” We then explain that, using the extrapolation of BER vs. power, an estimated $\sim 90 \mu\text{W}$ would be required to bring the BER down to 3.8×10^{-3} . (This estimate is based on the roughly $3 \times$ higher power need for PAM-4, which we derived from comparing the required eye opening to the OOK case.) We have included this numerical discussion so that the reader is fully aware of the limitations of our PAM-4 demonstration.
- Limiting factors for PAM-4: We discuss why the PAM-4 performance was limited. The main factor is highlighted: insufficient SNR, because the four-level modulation requires a higher SNR to distinguish the smaller differences between adjacent levels. The $\sim 75 \mu\text{W}$ maximum power resulted in only a modest SNR per level, leading to noise-induced symbol errors. Also, in the revised text, we write: “The PAM-4 signal is more sensitive to noise and to any transmitter nonlinearity, because the amplitude difference between adjacent symbols is much smaller than the ‘0’ to ‘1’ difference in OOK. In our system, the limited SNR (due to a maximum THz power of $75 \mu\text{W}$) and a slight nonlinearity in the QCL’s modulation response led to noticeable eye

closure and symbol-level distortion (Fig. 6d).” By highlighting this, we underscore the challenges faced in the PAM-4 experiment.

Overall, the revised manuscript now addresses the PAM-4 section appropriately: we acknowledge its limitations (the eye was not fully open, and BER exceeded the FEC limit). We have now incorporated this explanation into the revised manuscript of the Transmission performance of a THz FSO system for a **PAM-4 signal section (page 10, paragraph 2)**. We appreciate the reviewer’s suggestion to defer or remove the discussion of PAM-4 performance. However, we believe that with our additional analysis in the revised manuscript, the results will be of value to the reader and demonstrate the feasibility of the PAM-4 link in a system with a modest increase in THz power.

Comment 8. Can you provide any insight into the maximum achievable data rate of the system?

Response: We appreciate the reviewer for the question on the maximum achievable data of the system. Given the current 0.5 m free-space link using a 2.4 THz QCL transmitter and Schottky barrier diode receiver, the maximum data rate is fundamentally limited by the QCL’s modulation bandwidth (~5 GHz) and the available THz received optical power (~75 μ W). In practice, the system has demonstrated 4 Gbit/s using on-NRZ-OOK, which is near the limit for reliable operation. In this experiment, we used a 4 GHz bandwidth oscilloscope, which limited our testing to rates below 4 Gbit/s. However, as mentioned in the **Discussion section (page 12, paragraph 3)**, achieving higher rates or using higher-order modulation, such as PAM-4, would require substantially more bandwidth and signal power, as well as improved receiver sensitivity.

Reviewer #3 (Remarks to the Author):

The article shows results of THz transmission using a QCL, for 4 Gbit/s. The concept of this is not new, even if the data-rate is higher than SoTA for this frequency (carrier) and for the use of QCL. Also, it is difficult to assess the high impact nature of the work related to the main limitations observed in the results.

I have several comments that should be addressed to better assess the potential impact:

Response: We appreciate the reviewer's careful assessment of the work. While we agree that preliminary demonstrations of THz-QCL-based communications exist, we also note that the bandwidth demonstrated here is the first to surpass 1 Gbit/s and is over two orders of magnitude faster than previously reported in this frequency band. As such, this is the first successful demonstration of the THz QCL communication concept that delivers practically useful data rates. We note, for example, Reviewer 1’s statements that this “sets a clear milestone” and is a “highly impressive achievement”. Our step

change in performance is a combination of significant improvements in THz QCL packaging, Schottky-barrier diode technology, and in end-to-end system design. We have addressed the reviewer's constructive comments and questions below and believe these have improved the manuscript.

Comment: This is not clear what are the main limiting factors in the transmission, the source, the receiver, or a combination of the 2? As the title is emphasizing the THz laser, one could think the receiver is not the limiting part of the system, but it should be proven and if not, authors should explain what are the induced limitations from the receiver used. As simple example, the bandwidth and linearity of the Schottky diode used at receiver should be checked .

Response: We appreciate the reviewer for raising this concern, and we have addressed it directly in the Discussion. In summary, both the source and detector impose limitations, but in somewhat different ways:

- THz QCL (Source): The QCL is primarily limited in output power and modulation bandwidth. In our demonstration, the QCL's power (Received optical power: 75 μ W) was sufficient to achieve low BER at 4 Gb/s OOK over 0.5 m. For any attempt at higher data rates, longer distance, or more complex modulation (like PAM-4), the lack of additional power headroom becomes the critical factor (as we saw with PAM-4). Additionally, the QCL's bandwidth (we measured \sim 5 GHz 3 dB bandwidth in the link) limited the maximum symbol rate we could effectively use. This bandwidth is influenced by the QCL's impedance and packaging. The RF amplifier on the transmitter was 4.2 GHz, which also capped the bandwidth. So, the source is a primary limiting factor in both power and speed. We emphasize that improved QCL designs with higher power output, potentially orders of magnitude more, in the mW range, and higher RF modulation bandwidth would directly translate to improved performance, longer reach, and higher data rates.
- Schottky Detector (Receiver): The detector is limited mainly by its sensitivity and noise. The receiver system NEP is approximately 35 pW/Hz^{0.5}, which, while high for room-temperature THz detectors with GHz bandwidth, means that for our data rates, we require tens of microwatts of received power to achieve an acceptable SNR. A more sensitive detector (with a lower noise equivalent power, NEP) or one with gain (such as a THz heterodyne receiver or a bolometric detector with an amplifier) could significantly reduce the required transmitter power. In terms of bandwidth, our Schottky detector and subsequent electronics had sufficient bandwidth (\sim 20 GHz RF amplifier, 26.5 GHz bias-tee) that we did not fully utilize, as the 5 GHz system limit came more from the QCL side. Thus, the detector was not the limiting factor for reaching speeds of up to 4 Gbit/s; it can likely handle even faster data rates. However, it limits sensitivity: it essentially sets the minimum power required for a given BER. We explain this in the text, noting that lower-noise or more efficient detectors could improve the link margin.

We added these explanations in the revised manuscript to conclude in the **Discussion section (page 12, paragraph 3)** that both the source and detector need advancement for truly high-performance THz links. We explicitly write: “To significantly enhance THz link performance, improvements on both ends are required: higher-power, broadband THz QCLs and more sensitive, low-noise THz receivers.” We then discuss which factor was dominant in our case, specifically for OOK at 4 Gb/s and PAM-4 at 2 Gb/s, the transmitter power and linearity were clearly the limiting factors, as the detector could theoretically handle more power if available. We state this explicitly: “The NRZ-OOK at 4 Gbit/s and PAM-4 experiment was primarily limited by insufficient transmitter power.”

Comment: Related to the source, what about the radiation of the QCL signal in the free-space? A proper radiation with no echos is mandatory for telecom applications, was this checked?

Response: In our setup, the QCL output was carefully collimated and refocused to ensure that the received signal was primarily from the direct free-space path. Specifically, the QCL emission was collected using a 90° off-axis parabolic mirror to produce a well-collimated THz beam, and a second parabolic mirror focused the beam onto the Schottky diode’s horn antenna (with an aperture of approximately 0.38 mm). This configuration minimizes reflections and side lobes: the mirrors define a single optical path, while the cryostat and optical mounts were designed to reduce reflections. Moreover, the link length was short (0.5 m), and we did not detect evidence of multipath interference or echoes in the received waveform. The BER curves (Fig. 5) showed a consistent dependence on received power, indicating a single free-space path rather than multiple delayed signals.

We have now added a sentence in the **Methods: Receiver system section (page 14, paragraph 1)** to clarify that the optical path was collimated with matched off-axis parabolic mirrors and that no multipath effects were observed in the received signal.

For telecom applications over longer distances, additional measures such as anechoic environments or spatial filtering would be necessary. However, for this proof-of-principle 0.5 m demonstration, the radiation can be considered effectively echo-free.

Comment: For the BER evaluations, how was this done? When using an ON/OFF modulation, this is common to measure the BER with a bit error rate tester instead of using off-line demodulation. The BER values dependence with sequence length was checked?

Response: We thank the reviewer for this question about BER estimation. We have addressed our BER computation procedure in the **Methods: Offline digital signal processing and BER estimation section (page 15, paragraph 2)** in the revised manuscript:

- We used a pseudorandom binary sequence (PRBS) of length $2^{15} - 1$ (32,767 bits) as the test pattern for both NRZ-OOK and PAM-4. This PRBS was generated and loaded into the AWG,

and the same sequence was used to check errors at the receiver. As we have considered a long PRBS sequence length, we did not explicitly check the dependence of the sequence length on BER values.

- We clarify the sample size for BER calculation: we captured ~ 2 ms of data on the oscilloscope for each measurement. In the manuscript, we now mention: “A total of 1.98×10^6 bits were received and evaluated for each BER data point (corresponding to a 2 ms capture).” We note that each BER value in Fig. 5 is based on roughly 2 million bits.
- Error counting method: We performed direct error counting by comparing the transmitted and received bit sequences bit-by-bit. To do this, we first had to synchronize the sequences. We describe our synchronization algorithm, which we implemented offline. We aligned the known PRBS (transmitted sequence) with the received bit sequence and calculated the BER for each alignment, choosing the one with the fewest errors as the correct one. We mention that we also optimized the sampling instant and decision threshold by scanning over a small range around the initial estimates (as described in our **Methods: Offline digital signal processing and BER estimation section**). This yielded the minimum BER for the captured waveform.

Comment: Related to the linearity of the system, for the QCL part, I suggest to authors to analyze the linearity of the modulation of the QCL prior to experiments, to check the linearity level and identify in case the sources of no-linearity .

Especially, the signal in PAM-4 is (eye pattern) fully closed. Thus, the source

Response: We acknowledge the reviewer’s concern about the system linearity and PAM-4 transmission. We have added a dedicated discussion about system linearity and PAM-4 transmission. We identified that the transmitter (QCL) likely impacted the PAM-4 signal quality. We elaborate on this in the Discussion:

- The QCL, when directly modulated, has an output power that is a nonlinear function of the drive current, especially near threshold. For NRZ-OOK, a slight nonlinearity mainly affects the exact amplitude of the “1” level but does not cause inter-symbol distortion - it means the on-off modulation depth might not be 100%. This will have a negligible effect on OOK performance, since we only care about distinguishing between on and off (which we could still do reliably).
- For PAM-4, however, linearity is crucial because the information is in the amplitude levels. If the QCL’s output vs input is not perfectly linear, the intended equally spaced drive voltages will not produce equally spaced optical power levels. We understand that this occurred in our case: the middle levels, “01” and “10”, may have been compressed, closer together in power than intended, due to the QCL’s gain saturation near the bias point. This would make the eye openings uneven (which is observed). We reference this evidence by

pointing to the distorted eye in Fig. 6d and noting it in the text. However, as we discuss above, the primary limitation for PAM-4 modulation is insufficient power rather than the spacing of the levels.

- The Schottky diode detector also has a nonlinear transfer function, approximately quadratic I-V for zero-bias diode detection. However, under small-signal conditions, the diode's output voltage is proportional to input power, which in turn is proportional to the square of the E-field. For the OOK signals, the diode acts as a square-law detector, resulting in error-free transmission. For PAM-4, a square-law detector can, in principle, preserve the relative power levels since it's monotonic, but if the input range is large, the diode might enter a saturation region. However, we believe we stayed within the linear range of the detector as our input was at most tens of μW , which for these diodes is still in the linear detection regime. Hence, we determine the nonlinearity to derive from the source rather than the detector.

We revised the manuscript, incorporating these observations into both the **Results: Transmission performance of THz FSO system for PAM-4 signal section (page 10, paragraph 2)** and the **Discussion section (page 13, paragraph 3)**.

Comment: There is some mistake: authors discussed about “GBaud/s”, but this unit doesn't exist: “Baud” means the number of symbols per second, thus doesn't need to add “/s”.

Response: We appreciate the reviewer for pointing out this mistake. This has now been corrected in the revised manuscript.

We hope this response addresses the reviewer's queries. We once again thank the four reviewers for their professional comments and suggestions.

Yours sincerely,

Jayaprasath Elumalai (Email: j.elumalai@leeds.ac.uk) and

Joshua R. Freeman (Email: j.r.freeman@leeds.ac.uk)

School of Electronic and Electrical Engineering

University of Leeds, Leeds LS2 9JT, UK

(On behalf of all the co-authors)

Point-by-point responses to the reviewers' comments

Dear Reviewers,

Thank you for your comments concerning our manuscript entitled “Free-space Optical Communications at 4 Gbit/s Data Rate with a Terahertz Laser” (Manuscript ID: COMMSPHYS-25-0640A). We are pleased that Reviewers 1 and 2 are happy with the modification we have made and feel the manuscript is ready for publication.

We have carefully reviewed the comments of Reviewer 3 and respond to these below. Where needed, we have made further modifications to the manuscript, which are highlighted in the attached version.

Reviewer #3 (Remarks to the Author):

Authors did corrections, but some points remains unclear in my perspective.

Comment: For PAM4, I don't see really the interest here as the signal quality is very low and the eye pattern closed. It would be worth to determine the required power margin to make the system running, and discuss how it could be achieved (or not) using the QCL approach? .

Response: We appreciate the reviewer's concern, and while we agree that the eye is closed and does not constitute successful data transmission, it does allow us to draw useful conclusions in the manuscript, including determining the power required for successful PAM-4 transmission. We have clarified the second paragraph of section “Transmission performance of THz FSO system for PAM-4 signal” to ensure these points are clear:

In our measured PAM-4 eye (Fig. 6d), the eye openings are partially closed and uneven, indicating significant noise and some level of distortion. ... the eye diagram (Fig. 6d) should show three eye openings corresponding to the four symbol levels. Compared to NRZ-OOK, where only one eye opening is required, this suggests that in our direct-detection system approximately three times more THz power would be required with PAM-4 to achieve the same BER at the same symbol rate and noise bandwidth. The NRZ-OOK at a data rate of 1 Gbit/s (1 Gbaud) required 30 μ W of optical power to reach the HD-FEC BER threshold. From this, we estimate that $\sim 90 \mu$ W would be required to bring the BER down to 3.8×10^{-3} for 2 Gbit/s PAM-4. Consequently, at the maximum received power of 75 μ W that we can achieve in this experiment, the BER for PAM-4 is expected to be above the HD-FEC threshold, and hence low error-rate communication would not be possible, even if FEC were employed.

Furthermore, from the eye diagram shown in Fig. 6d it is clear that there is some non-uniform spacing between the levels, allowing us to observe (same paragraph):

The lowest and highest levels (corresponding to symbols “00” vs “11”) are furthest apart and most distinguishable, while the intermediate levels (“01” and “10”) have smaller separations and are more prone to overlap due to noise.

We have also modified the Discussion (second paragraph) to clarify how PAM-4 could be achieved:

Our results show that while PAM-4 offers higher spectral efficiency than OOK, it also introduces greater sensitivity to noise and amplitude distortion. Successful PAM-4 data transmission could be achieved with a QCL by increasing the modulated power (from 75 μ W to 90 μ W in this case), increasing receiver sensitivity, and implementing digital signal processing techniques, such as equalization, to mitigate distortions. Higher power QCL sources could be achieved by the use of larger laser ridges, optimized waveguide extraction efficiency, or more efficient active region designs.

We believe this explains our motivation for including the PAM-4 data. It allows us to estimate the required optical power at the receiver to use this multi-level format, and it also highlights that nonlinearities in the QCL response must be avoided for efficient multi-level encoding.

Comment: About BER curves, something is surprising: when increase the data-rate by a factor of 4, the required power level remains more or less constant. This is not expected from physics.

Response We appreciate the reviewer’s concern on this point and agree it is not what you would expect from the Additive White Gaussian Noise (AWGN) model. In that ideal case, when the noise is assumed to be constant across all frequencies and the receiver’s equivalent noise bandwidth is assumed to scale linearly with symbol rate, increasing the data rate by a factor 4 theoretically increases the integrated noise by 4, and a constant BER would require 4 times more electrical signal power (maintaining the electrical SNR at a constant value). In our case, however the noise is not flat: The measured noise PSD (Supplementary Fig. S3) rolls off with frequency. Most noise power lies below \sim 1-1.5 GHz, and above \sim 2-3 GHz as the PSD declines. To highlight this, we have modified the text (**section “Transmission performance analysis of the THz FSO communication system”, page 9, paragraph 3**):

In an AWGN system, the theoretical expectation is that an increase of 3 dB THz received optical power, leading to a 4-times increase in electrical signal power from our direct detection system, is expected for a four times change in the data rate while maintaining the same BER in an OOK system. However, our observed increase from 1 to 4 Gbit/s (+1.2 dB per four times increase) is somewhat more modest, likely because other factors, such as a slightly suboptimal threshold determination and the measured noise power spectral density of the detector reducing as the frequency is increased (Supplementary Information Section 2).

Comment: Authors said that they expect 3 dB extra power when multiply the data-rate by 4. This is somehow strange as when increasing the data-rate by 4, the spectrum frequency extension is also

increased by 4. Thus integrated noise bandwidth is increased by 4, which is 6 dB more, and to compensate this, an extra 6dB power is required. Can authors discussed their 3 dB prediction?

This comment refers to section “Transmission performance analysis of the THz FSO communication system”, paragraph 3, discussed above in response to the previous comment.

The 3dB increase in power we are referring to is the received optical (Terahertz) power, arriving at the detector. In this work we are using the Schottky barrier diode as a ‘direct’ or ‘envelope’ detector, which means that the voltage produced by this detector is proportional to optical/THz power arriving (we measure a responsivity of 31V/W). When the voltage from the detector is increased by a factor of 2, the power of the modulated electrical signal passed (via the LNA) to the oscilloscope will increase by a factor 4. Hence, when the data rate is increased by a factor 4 one would expect a 4-times increase in noise (assuming AWGN and noise bandwidth increased by a factor of 4), and the required electrical power must be increased by a factor 4 to achieve the same electrical SNR, but to achieve this the required optical power needs to be increased only by a factor 2 (3dB) in this direct-detection system.

We have clarified this point in the manuscript, modifying the sentence (**page 9, paragraph 3**):

In an AWGN system, the theoretical expectation is that an increase of 3 dB THz received optical power, leading to a 4-times increase in electrical signal power from our direct detector, is expected for a four times change in the data rate while maintaining the same BER in an OOK system.

Comment: End of page 6, there is a mistake : « ... a low-noise amplifier with +15dB gain to improve the signal-to-noise ratio”. This is not correct: any amplifier degrades the SNR. The LNA increases the signal to make it sufficiently powerfull for the receiver, but degrades the SNR of the signal in any case..

Response: We appreciate the reviewer for pointing this out. We have now clarified this in the revised manuscript in the **Results section (page 6, paragraph 1)**:

The demodulated message is amplified using a low-noise amplifier with +15 dB gain to provide sufficient voltage swing for the digitizer and to overcome the ADC front-end noise.

We hope this response addresses the reviewer’s queries. We once again thank the three reviewers for their professional comments and suggestions.

Yours sincerely,

Jayaprasath Elumalai (Email: j.elumalai@leeds.ac.uk) and

Joshua R. Freeman (Email: j.r.freeman@leeds.ac.uk)

School of Electronic and Electrical Engineering

University of Leeds, Leeds LS2 9JT, UK

(On behalf of all the co-authors)

Dear Reviewers,

Thank you for your comments concerning our manuscript entitled “Free-space Optical Communications at 4 Gbit/s Data Rate with a Terahertz Laser” (Manuscript ID: COMMSPHYS-25-0640B).

We are pleased that Reviewers 1, 2, and 3 are happy with the modification we have made and feel the manuscript is ready for publication.